# Characterization of Shock Wave Damages in Explosion Welded Mo/Cu Clads

**Pradeep Kumar Parchuri [1], Shota Kotegawa [1], Kazuhiro Ito [1,*], Hajime Yamamoto [1], Akihisa Mori [2], Shigeru Tanaka [3] and Kazuyuki Hokamoto [3]**

[1] Joining and Welding Research Institute, Osaka University, 11-1 Mihogaoka, Ibaraki, Osaka 567-0047, Japan; pradeepkumar.rut@gmail.com (P.K.P.); tennisteam0614@yahoo.co.jp (S.K.); h.yamamoto@jwri.osaka-u.ac.jp (H.Y.)

[2] Department of Mechanical Engineering, Sojo University, 4-22-1 Ikeda Nishi-ku, Kumamoto, Kumamoto 860-0082, Japan; makihisa@mec.sojo-u.ac.jp

[3] Institute of Pulsed Power Science, Kumamoto University, 2-39-1 Kurokami, Chuo-ku, Kumamoto, Kumamoto 860-8555, Japan; tanaka@mech.kumamoto-u.ac.jp (S.T.); hokamoto@mech.kumamoto-u.ac.jp (K.H.)

\* Correspondence: ito@jwri.osaka-u.ac.jp; Tel.: +81-6-6879-8659

**Abstract:** The shock wave damage during explosive welding has not been reported in a flyer Mo plate of the Mo/Cu clads. However, it would be an inevitable problem in group VI elements. This study was aimed to characterize the shock wave damage in the Mo plate, that is less brittle than a W plate, of explosive welded Mo/Cu clads. Cladding at low horizontal collision velocities leading to high collision angles was expected to enhance the shock wave damage, and the clads resulted in less elongation in bending tests. On the other hand, in the clads obtained at high horizontal collision velocities (HCVs) with low collision angles, their bending elongation increased significantly. The shock wave damage penetrated from the surface of a Mo plate to the Mo/Cu interface, and thus reducing thickness of a Mo plate of bending specimens increased bending plastic strain. The shock wave damage is associated with kinetic energy imparted to the flyer Mo plate, and thus loss of kinetic energy due to formation of an intermediate layer at the interface and reducing thickness of a flyer Mo plate would be very helpful for decrease of shock wave damage.

**Keywords:** Mo/Cu clads; explosive welding; shock wave damage; micro cracks; bending elongation; intermediate layer formation

## 1. Introduction

The dissimilar Mo/Cu clads can be potential materials in applications such as heat sinks [1] due to high melting point, superior wear resistance, high sputtering resistance against plasma and low vapor pressure of a Mo plate, and high thermal conductivity of a Cu plate. Especially dissimilar Mo/Cu clads can serve as plasma facing materials in thermonuclear fusion energy applications [2], which is an important source of energy in the near future [3]. On the other hand, large differences in the thermal and physical properties of Mo and Cu (Table 1 [4]) make it difficult to join them using conventional fusion welding techniques [1]. Ion implantation and direct laser cladding were shown to be ineffective to obtain successful bonding between Mo and Cu plates. Although diffusion bonding technique could produce high strength Mo/Cu joints, the Ni-interlayer usage was essential [5–7]. Explosive welding is an impressive solid state welding technique, proven to be able to weld directly dissimilar metals together. The combination of dissimilar metals is difficult to weld by any other existing welding techniques [8–15]. In the explosive welding, the detonation of high energy explosive accelerates one of the plates (a flyer plate) to be joined to collide against the other plate (a base plate). Consequently, pressure generated at the collision interface exceeds the dynamic yield strength of both the metal plates and metal jets were emitted from the collision interface and the oxide scale and contaminations on the joining

surfaces can be scraped off by the jetting; the atoms of the topmost layers of the joining surfaces could come into intimate contact, resulting in formation of strong metallurgical bond [16].

**Table 1.** Thermo-physical properties of Mo and Cu plates [4].

| Materials | Mo | Cu |
|---|---|---|
| Melting temperature (K) | 2896 | 1356 |
| Density (g/cm$^3$) | 10.2 | 8.96 |
| Thermal conductivity (Wm$^{-1}$K$^{-1}$) at 300 K | 142 | 398 |

Several studies have been reported on explosive cladding between Mo and Cu plates. A detailed study on bonding mechanism and microstructural features around the interface were reported. A bond layer with a thickness of about 10 μm existed at the interface in the immiscible binary Mo–Cu phase diagram. It was recommended to obtain Mo/Cu clads at impact velocity as low as possible in order to avoid cracks in the flyer Mo plate [6]. Manikandan et al. employed underwater shock waves to obtain explosive welded Mo/Cu clads [17]. Their study predominantly focused on obtaining a wavy interface since the wavy interface is believed to exhibit high strength for any welds produced by the explosive welding [18]. Furthermore, they suggested that cladding at right lower limit in the weldability window could assure a sound Mo/Cu clad. Noted that no one reported the mechanical properties of the as-explosive-welded Mo/Cu clads. On the other hand, an intermediate layer at the interface was proven to be beneficial over a wavy interface in the immiscible binary systems such as Nb–Cu and Ta–Cu binary systems [19]. In addition, the effect of shock wave damage has not been raised and discussed in the as-explosive-welded Mo/Cu clads. The shock wave damage causes a significant deterioration in a flyer brittle metal such as W and Mo of as-explosive-welded clads. Parchuri et al. [20] demonstrated that the bending strength and elongation of as-explosive-welded W/Cu clads was significantly dependent on damage amount of the flyer W plate. This suggests that there is a necessity to investigate the mechanical properties of the as-explosive-welded Mo/Cu clads and to understand relationship between the shock wave damage and explosive welding conditions, that can control mechanical properties of the Mo/Cu clads.

This study aimed to characterize shock wave damage, e.g., using indentation test on cross sections of the flyer Mo plate of the explosive welded Mo/Cu clads and to identify influence of the explosive welding condition on the shock wave damage. The clads were produced at low and high horizontal collision velocities (LCV and HCV, respectively, here after LCV and HCV clads. The HCV maintains a low collision angle, that is consistent of the right lower limit in the weldability window. In contrast, the LCV gives a high collision angle. Cladding at HCV resulted in crack-free flyer W plates [20], and less shock wave damage in the flyer Mo plate of the HCV clads would be expected. This study intended to elucidate the shock wave damage related to explosive welding conditions.

## 2. Experimental Details

The Mo/Cu LCV and HCV1–4 clads were obtained at various horizontal collision velocities, and experimental setups and setting parameters used to obtain their clads are shown in Figure 1a,b, respectively. The LCV clad was obtained using a gelatin medium (Figure 1a), and the theoretical aspects of the experimental setup are described elsewhere [21]. The ammonium nitrate fuel oil (ANFO)-A, an ANFO-based explosive (explosive characteristics were the same as mentioned in [21]), was employed to clad a flyer Mo plate to a Cu plate separated by a stand-off distance (0.5 mm). The commercially available Mo plate (from the Nilaco Corp. City, Japan). with hardness of 259–263 HV was used and the sample size for explosion welding was 50 mm square area and 0.5 mm thickness. All the HCV clads were obtained using water as medium (Figure 1b), and the theoretical background of the experimental setup is discussed elsewhere [22]. A SEP-based explosive produced by Asahi-Kasei Chemicals Corp (City, Japan) (explosive characteristics are the same as mentioned

in [22]) was employed to clad a flyer Mo plate to a Cu plate separated by a stand-off distance (0.1 mm) for the HCV1–4 clads. The explosive was put on polymethyl methacrylate plate, and its thickness ($d_{1E}$ and $d_{2E}$), length ($l_E$), and inclination angle ($\alpha$) was changed to control the imparted kinetic energy of the flyer Mo plate, as shown in Table 2. Similarly, thickness of the flyer Mo plate was changed. Noted that two different kind mediums were used to obtain the required samples for this study, while it was confirmed that little difference was detected in microstructure and mechanical properties between the samples prepared with the two mediums [21]. All the explosive welding experiments were conducted at the Institute of Pulsed Power Science, Kumamoto University, Japan, and the explosives were supplied by Kayaku Japan Co. It is to be mentioned that, the base copper plate used in LCV and HCV1 clads was pure copper, whereas it included 5 at.% of Sn in HCV2-4 clads against our will. The stress in stress–strain curves of the HCV2-4 clads was higher than those of LCV and HCV1 clads, however the difference did not affect the goal of this study.

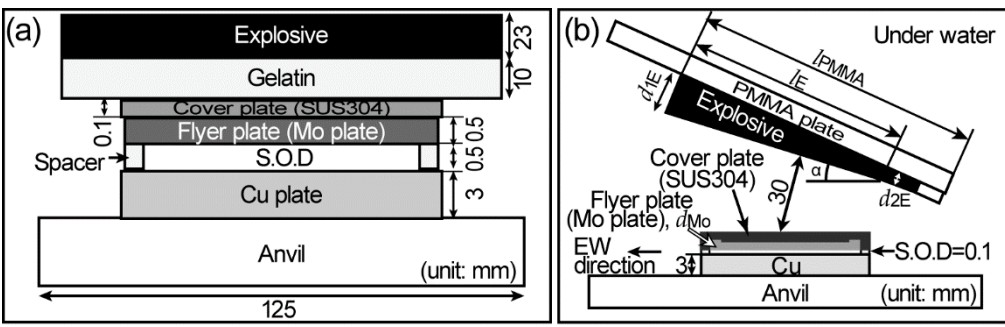

**Figure 1.** Schematic illustrations of explosive welding settings used for (**a**) low horizontal collision velocity (LCV), (**b**), high horizontal collision velocity (HCV)1–4 clads. (**a**,**b**) S.O.D: stand-off distance, (**b**) PMMA: polymethyl methacrylate, and setup parameters for HCV1–4 clades are shown in Table 2.

**Table 2.** Under-water explosion welding setup parameters for HCV1–4 clads.

| Clads | Mo Plate Thickness $d_{Mo}$ (mm) | Explosive Thickness $d_{1E}$ (mm) | Explosive Thickness $d_{2E}$ (mm) | Explosive Length $l_E$ (mm) | PMMA Plate Length $l_{PMMA}$ (mm) | Explosive Inclination Angle $\alpha$ (°) |
|-------|------|------|---|-------|-------|------|
| HCV1 | 0.5 | 13.8 | 5 | 101.3 | 173.6 | 15 |
| HCV2 | 0.5 | 11.2 | 3 | 113.7 | 175.5 | 17.5 |
| HCV3 | 0.5 | 13.8 | 5 | 101.3 | 173.6 | 17.5 |
| HCV4 | 0.3 | 13.8 | 5 | 101.3 | 173.6 | 17.5 |

The HCV2 clad was produced with explosive setting parameters described in blue different from those of the HCV1, 3 and 4 clads described in black, and a thin Mo plate of 0.3 mm thickness, described in red, was used in the HCV4 clad in Figure 1b. Tilting angles of the explosive for the HCV clads were also described in Figure 1b. The weldability window for the explosive welded Mo/Cu clads together with contour curves of the estimated kinetic energies imparted to the flyer Mo plates were illustrated in Figure 2. The thinner Mo plate in the HCV4 clad (a red broken curve) than those in the other HCV clads (black solid curves) decreased the imparted kinetic energy. The kinetic-energy contour curves calculated according to the thicknesses of flyer Mo plates are shown in Figure 2 with the black curves for the LCV and HCV1, 2 and 3 clads, and red broken curves for the HCV4 clad.

Cross-sectional observation of the as-explosive-welded specimens were carried out using an optical microscope (OM) and a field emission scanning electron microscope (FE-SEM). Inverse pole figure (IPF) and grain reference orientation deviation (GROD) maps were obtained using another FE-SEM equipped with an electron back-scattered diffraction (EBSD) camera. Scanning ion microscope (SIM) technique was employed to detect the presence of micro defects in cross-sections of Mo plates of the as- explosive-welded clads.

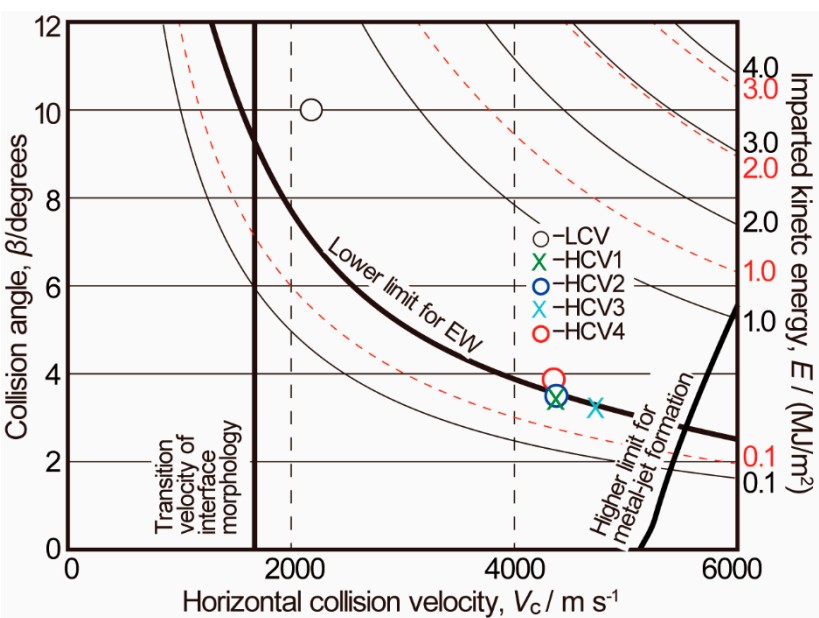

**Figure 2.** Relationship between estimated collision angles and horizontal collision velocities for LCV and HCV1–4 clads, together with contour curves of the estimated kinetic energies imparted to a flyer Mo using red broken curves for the HCV4 clad and black curves for the other clads.

Three-point bending tests were carried out at the crosshead speed of 0.5 mm/m, the punch tool diameter of 6 mm, and at room temperature to evaluate the mechanical properties of the LCV clad. Similarly, systematic three-point bending tests were carried out to investigate the shock wave damage in the Mo plates of the HCV2 and HCV4 clads. Three sets of Mo/Cu bending specimens were prepared by polishing both Mo and Cu plates of the clads to the required thickness for keeping a ratio of Mo- and Cu-plate thicknesses. The specifications of the bending test specimens were mentioned in Table 3, and bending load was applied on the Cu plate side of the specimens. The bending strain $\varepsilon$ was estimated from the crosshead displacement in the tensile testing machine (Shimadzu Corp.) based on the common conversion Equation (1);

$$\varepsilon = 6t/d^2 \times (crosshead\ displacement),\tag{1}$$

and the bending stress $\sigma$ was estimated by the Equation (2);

$$\sigma = 3d/(2\,w\,t_{clad}^2) \times (load).\tag{2}$$

**Table 3.** Dimensions ($t$, $w$) of three-point bending test specimens and fulcrum distances ($d$) in the specimens for HCV2 and HCV4 clads: $d = 3t + \varphi$ ($\varphi = 6$ mm).

| | Thicknesses, $t$/mm | | Widths, $w$/mm | | Fulcrum Distances, |
| Clads | Cu Plate | Mo Plate | HCV2 | HCV4 | $d$/mm |
|---|---|---|---|---|---|
| 2.5 | 2.3 | 0.2 | 3.98 | 4.09 | 13.5 |
| 2.0 | 1.84 | 0.16 | 4.81 | 4.98 | 12 |
| 3.15 | 2.9 | 0.25 | 5.95 | 5.86 | 15.45 |

The bending stress is usually larger than the tensile/compressive stress, while the comparison of strain between the bending and tensile tests is a little bit complicated.

Vickers hardness tests were carried out on the Mo cross sections of clads at room temperature with an applied load of 0.098 N, loading time of 30 s, and interval distance of 0.05 mm to determine the variation of hardness with distance from the interface for the as-explosive-welded HCV2 and HCV4 clads. In addition, to investigate the effect of shock

wave damage, Vickers hardness tests were conducted on the Mo cross sections of clads at room temperature with an applied load of 0.49 N, to obtain relatively large apparent contact area probably including invisible micro defects formed by the shock wave damage, loading time of 30 s, and interval distance of 0.05 mm parallel to the interface at three positions away from the interface of the Mo plates.

## 3. Results and Discussions

The as-explosive-welded Mo/Cu clads revealed sound joining and absence of surface crack or defect on a Mo plate of all the clads. Cross-sectional FE-SEM images of all the as-explosive-welded Mo/Cu clads revealed wavy interface, suggesting sound joining, in all the clads as shown in Figure 3. However, cracks were observed at the interface of only the HCV1 and HCV3 clads. They propagated along the interface and penetrated through a Mo plate (Figure 3b,d). These two estimated explosive welding conditions are almost close to the lower limit for explosive welding (Figure 2), and in fact the conditions probably became below the limit. On the other hand, the clads obtained at LCV, HCV2 and HCV4 revealed sound bonding, and their cross-sections were free from any visible cracks. Thus, those three clads were considered for further investigation to reach the goal of this study. The wave period of an undulant interface was five or more times longer in the LCV clads than that in the HCV clads.

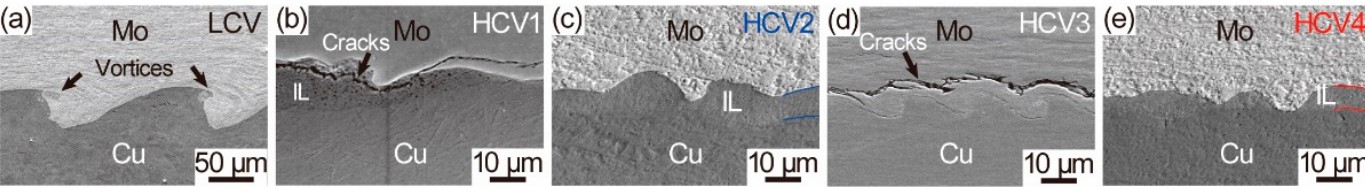

**Figure 3.** Cross-sectional field emission scanning electron microscope (FE-SEM) images of as-explosive-welded clads produced at (**a**) LCV, (**b**) HCV1, (**c**) HCV2, (**d**) HCV3, and (**e**) HCV4. (**b**,**c**,**e**) IL: Intermediate layer.

### 3.1. LCV Clad

Cross-sectional FE-SEM image of the LCV clad exhibited a continuous wavy interface with vortices at a wave crest (Figure 3a), indicating generation of metal jets. The estimated collision angle of the LCV clad was much higher than those of HCV clads (Figure 2), and thus the high velocity jets generated at the interface escaped easily during bonding, while a small amount of jets was trapped leading to formation of vortices at a wave crest. When a Mo plate hit a Cu plate with high impact velocity at an oblique angle, severe shear deformation took place at the collision interface and resulted in formation of an asymmetric wavy interface with distance between trough and neighboring crest in the welding direction (from right to left) much less than half of the total wavelength due to the density difference between Cu and Mo [23–26]. Similarly, the features of wavy interface depended on difference in the thermo-physical properties of the base metals such as melting point, density, and thermal conductivity. Furthermore, the base plates at the interface seemed to be elongated in the welding direction, meaning fluid like behavior of the materials at the colliding interface, caused by high strain rates and high temperatures generated at the collision point during explosive welding [27].

A three-point bending stress–strain curve of the LCV clad was obtained at a cross head speed of 0.5 mm/m at room temperature as shown in Figure 4a. The bending strength obtained in the LCV clad was higher than that of a pure Cu plate, suggesting a sound joining. The bending stress increased with increasing bending strain, leading to about 9% plastic strain. Crack was initiated at the surface of the Mo plate, and penetrated through the Mo plate and then propagated along the wavy interface without failure of the Cu plate (Figure 4b,c). The limited bending plastic strain seems to be caused by shock wave damage in the Mo plate due to high kinetic energy imparted to a flyer Mo plate during explosive welding [19,20]. Noted that LCV clad was obtained at higher kinetic energy than HCV

clads (Figure 2). This suggests that employing the LCV condition is not recommended for obtaining explosive welded Mo/Cu clads, leading to employing the HCV conditions making the imparted kinetic energy as low as possible at lower collision angles.

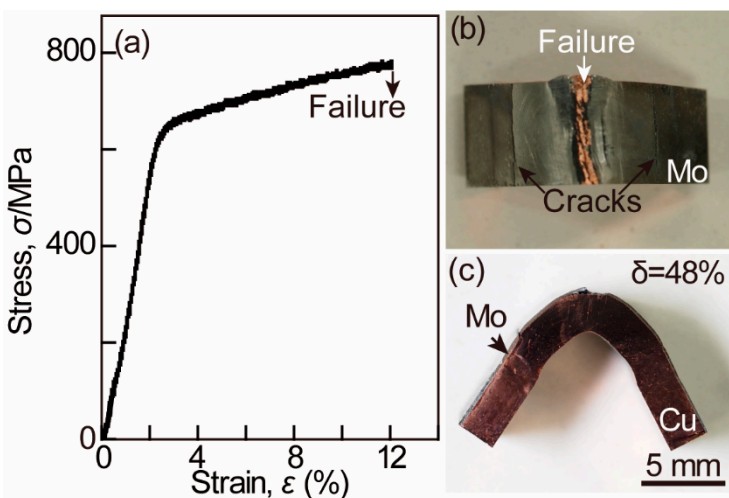

**Figure 4.** (**a**) A bending stress–strain curve at the cross-head speed of 0.5 mm/m at room temperature. (**b**) Top-view and (**c**) cross-sectional optical microscope (OM) images of a Mo/Cu LCV specimen after a bending test.

*3.2. HCV Clads*

Cross-sectional FE-SEM images of both the HCV2 and HCV4 clads exhibited a continuous intermediate layer at the interface partitioned by blue and red curves (at the right-hand edge), respectively, as shown in Figure 3c,e. This is associated with the high collision velocities and low collision angles employed to obtain those clads. The high velocity jets generated at the interface could be trapped at low collision angles, resulting in a continuous intermediate layer at the interface [28]. The average composition of the intermediate layers for both the HCV clads was found to be almost similar, and the contents of Mo, Sn, and Cu were found to be about 16, 5 and 79 at.%, respectively. The intermediate layer of the HCV4 clad was characterized to be a little thinner than that of the HCV2 clad. This is associated with lower imparted kinetic energy in the HCV4 clad. The intermediate contrast of the intermediate layer at the interface was close to that of the Cu plate, associating with large amount of copper in the intermediate layers. Furthermore, the curvy Mo/Intermediate-layer and flat Intermediate-layer/Cu interfaces observed at the interface were caused by the density difference between Mo and Cu; the cumulative jets produced at the collision interface moved toward a Mo plate with higher density than toward a Cu plate, leading to deformation of the Mo plate, then trapped. Thus, the Mo/Intermediate-layer interface was wavier than the intermediate-layer/Cu interface. Figure 5 shows EBSD-IPS and -GROD maps obtained on cross sections of the HCV2 and HCV4 clads. The intermediate layer exhibited longitudinal growth grains in contrast to the lateral elongated rolling texture in the Mo plate and equiaxed-grain structure in the Cu plate (Figure 5a,c). The lateral elongated rolling texture was finer in the HCV4 clad than that in the HCV2 clad, since the Mo-plate thickness of the HCV4 clad was thinner than that of the HCV2 clad (Table 2). The intermediate layer was found to be divided into two regions of upper thin and lower thick layers. This suggests that the intermediate layer was melted at the collision and grains grew perpendicular to the interface from the upper Mo and lower Cu plane. Accordingly, the intermediate layer exhibited little strain, in contrast the Mo and Cu base plates exhibited large strains after the explosive welding (Figure 5b,d). Conductivity of the Cu plate is larger than that of the Mo plate, and the grain growth from the lower Cu plane was faster than that from the upper Mo plane.

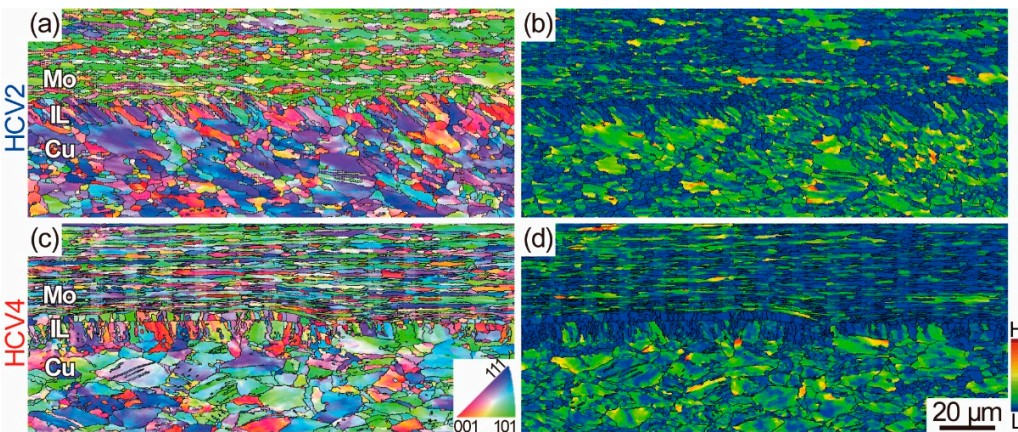

**Figure 5.** Cross-sectional (**a**,**c**) electron-back-scattered-diffraction (EBSD) - inverse-pole-figure (IPF) and (**b**,**d**) EBSD - grain-reference-orientation-deviation (GROD) maps of as-explosive-welded (**a**,**b**) HCV2 and (**c**,**d**) HCV4 clads. The intermediate layers (ILs) produced at the Mo/Cu interfaces.

The Mo plate was the hardest followed by the intermediate layer and Cu plate was the softest. As mentioned in the LCV clad, the mechanical property of the Mo plate after explosive welding is responsible for that of the clad. Vickers hardness variation on cross sections of the Mo plate with distance perpendicular to the interface in the as-explosive-welded HCV2 and HCV4 clads is shown in Figure 6. The hardness value decreased monotonously according to the distance from the interface in the HCV2 clad (Figure 6a). On the other hand, the value increased according to the distance from the interface until the center of Mo plate, and decreased according to the distance from the center in the HCV4 clad (Figure 6b). The hardness variation seems to depend on the Mo-plate thickness. This suggests that the trend difference in hardness change with respect to distance from the interface in the flyer Mo plate of as-explosive-welded Mo/Cu clads is associated with Mo-plate thickness. The lower hardness near the interface in the HCV4 clad is associated with heat generation due to intermediate-layer formation during explosive welding. In contrast, that in the HCV2 was the highest near the interface in the flyer Mo plate, although the HCV2 clad had also possessed an intermediate layer at the interface. The Mo-plate thickness of the HCV2 clad was about 2 times thicker than that of the HCV4 clad, and the hardness difference near the interfaces of the Mo plate in both the clads seemed to be due to the heat transfer difference. This is in good agreement with the lower plastic strain near the interface observed in the flyer Mo plate of the HCV4 clad than the HCV2 clad, as shown in Figure 5b,d. The hardness increase near the interface is associated with plastic deformation due to high horizontal collision velocity. In contrast, the hardness decrease near the interface was not associated with the micro defects, suggesting plastic strain would not be reduced. The maximum hardness was higher in the HCV4 clad (about 315 HV) than in the HCV2 clad (300 HV). The higher hardness in the HCV4 clad is associated with the finer rolling texture.

On the other hand, the heat generation due to the intermediate-layer formation leaded to loss of the imparted kinetic energy. This is expected to decrease the shock wave damage. However, the shock wave damage still existed in both the flyer Mo plate of the HCV2 and HCV4 clads, leading to hardness decrease near the flyer-plate surface due to micro defects such as cracks and voids. This resulted in decrease plastic strain in the flyer Mo plates as well as in the clads.

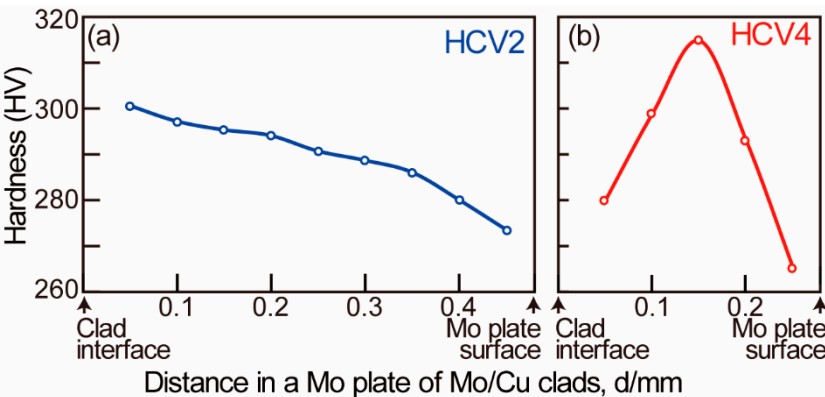

**Figure 6.** Vickers hardness profiles obtained at room temperature with an applied load of 0.098 N in the cross section of a Mo plate between the interface and Mo plate surface for (**a**) HCV2 and (**b**) HCV4 clads.

### 3.3. Shock Wave Damage in HCV Clads

To control the shock wave damage region for the bending tests, both the Mo and Cu plates of as-explosive-welded HCV2 and HCV4 clads were polished carefully keeping the ratio of Mo- and Cu-plate thickness the same as possible, as shown in Table 3. Figure 7 shows stress–strain curves of a series of the three specimens with different thicknesses, and top-view and cross-sectional OM images of the specimens after bending tests are shown in Figure 8. The HCV2 clads have always exhibited higher bending stress than the HCV4 clads (Figure 7), although the finer rolling texture and the higher maximum hardness in the HCV4 clads was observed (Figures 5 and 6). This is associated with a little thicker intermediate layer of the HCV2 clad than that of the HCV4 clad. On the other hand, the bending strain depended on the specimen thickness. The dependence was different in the HCV2 and HCV4 clads. Large bending strain could be observed at the thickness of 2.00 mm for the HCV2 specimens, while those could be observed at the thicknesses less than 2.50 mm for the HCV4 specimens. Some of the Mo plates in the specimens failed during bending tests (Figure 8a$_1$–a$_6$), while no failure was observed in the specimens with large bending strain of about 40% (Figure 8a$_7$–a$_{12}$). The initiation of cracks on surfaces of the Mo plates and their propagation toward the interface of the clads were observed (e.g., Figure 8a$_1$–a$_4$). Noted that bending strain of the clad specimens was associated with whether the damaged layer in the Mo plate of the as-explosive-welded clads could be removed or not. In addition, the shock wave damage was accumulated near the Mo plate surface of the as-explosive-welded clads than near the interface. This suggests that the shock wave damage caused by the explosive welding enhanced especially in the Mo plate of the as-explosive-welded Mo/Cu clads and the damage was accumulated at high density at the topmost Mo plate.

Shock wave damage is most commonly occurring problem in the explosive welding of brittle metals such as W. This is caused by the interaction between shock waves and reflected tensile waves, which propagate in the opposite directions each other during explosive welding. The shock waves generate at the collision interface caused by a part of imparted kinetic energy, while the reflected tensile waves generate at a flyer-plate surface caused by reflection of the shock waves. The shock wave damage is predominant in the brittle metal plates such as W and Mo with rolled texture that consists of layer structure stacking of elongated grains. Tensile stress produced by the shock and reflected tensile waves in the opposite directions was applied perpendicular to the layer boundaries, eventually leading to formation of lateral cracks along the boundary in a W plate of as-explosive-welded W/Cu clads [29,30]. The presence of shock wave damage in a flyer plate of explosive welded clads deteriorates its mechanical property, accordingly the mechanical property of clads can be deteriorated. To reduce the shock wave damage in a flyer plate as similar to that before explosive welding, loss of the imparted kinetic energy at the collision interface should be as

large as possible. Cladding at high horizontal collision velocity maintaining a low collision angle is suggested to deliver large kinetic-energy loss at the collision interface. In the similar explosive welding condition, almost no shock wave damaged flyer Nb or W plates could be cladded onto Cu plates, as reported in the earlier study [19,20].

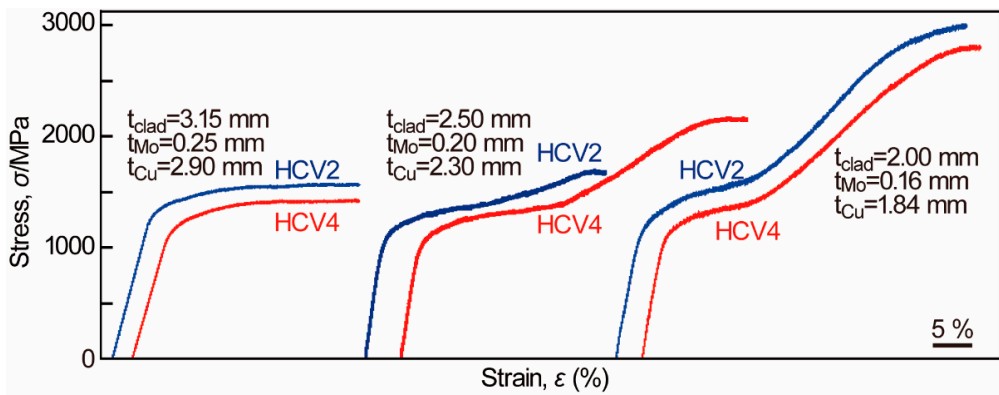

**Figure 7.** Bending stress–strain curves of HCV2 and HCV4 specimens at room temperature and 0.5 mm/m cross-head speed for three clad thicknesses of 3.15, 2.50 and 2.00 mm, keeping the ratio of Mo- and Cu-plate thickness the same as possible.

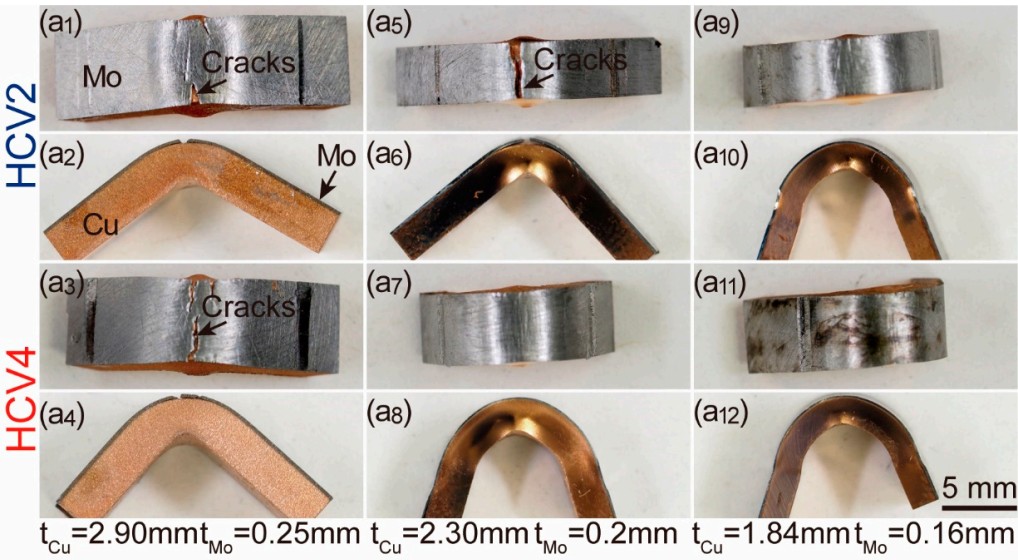

**Figure 8.** Top-view and cross-sectional OM images of the HCV2 and HCV4 clads after bending tests. (**a1**–**a6**) Some of the Mo plates in the specimens failed during bending tests, (**a7**–**a12**) while no failure was observed in the specimens with large bending strain of about 40%.

Although clear evidence of macro cracks could not be found in cross-sectional FE-SEM images of the flyer Mo plates in the as-explosive-welded clads in comparison to the as-explosive-welded W/Cu clads [20], micro cracks and voids around grain boundaries were found in the cross-sectional SIM images perpendicular to the Mo-plate surface obtained underneath the surface of the flyer Mo plates of the LCV, HCV2 and HCV4 clads (Figure 9). Noted that kinds of micro defects were limited underneath the surface of the flyer Mo plates and were hardly observed in any other regions. The invisible defects probably formed in the flyer Mo plate of the clads other than the topmost region. The shock wave damage was found around grain boundaries in a Mo plate with rolled texture. Thus, the grain boundaries in the flyer Mo plate seemed to be weakened, depending on the intensity and wavenumber of shock waves and reflected tensile waves. The shock wave damage region in a flyer Mo plate can vary with distance away from the interface.

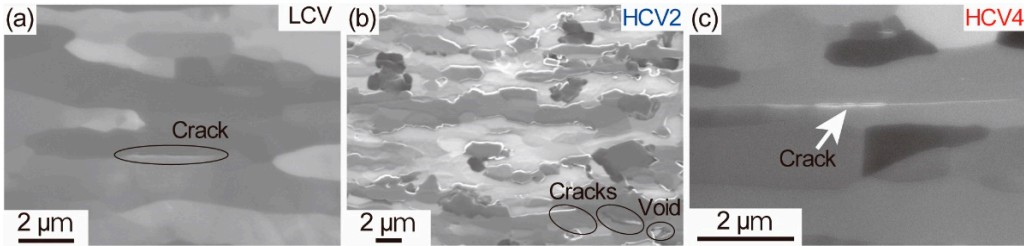

**Figure 9.** Cross-sectional SIM images near the Mo-plate surface of the as-explosive-welded (**a**) LCV, (**b**) HCV2 and (**c**) HCV4 clads.

Vickers hardness tests were conducted parallel to the interface at the three positions of 0.15, 0.20 and 0.25 mm away from the interface at room temperature with an applied load of relatively large 0.49 N load, to obtain relatively large apparent contact area probably including invisible micro defects formed by the shock wave damage, on cross sections of the flyer Mo plates of the as-explosive-welded HCV2 and HCV4 clads, as shown in Figure 10. The monotonic hardness decrease from the interface to the flyer Mo-plate surface was observed in the HCV2 clad (Figure 6a), while the maximum peak in hardness at the center of a flyer Mo plate was observed in the HCV4 clad (Figure 6b). The trend of hardness in the three positions was under monotonic decrease with increasing distance from the interface for both the MCV2 and MCV4 clads. The average hardness on the flyer Mo plate was different in different positions for the HCV4 clads in comparison to the same hardness at all positions in the HCV2 clads. The average hardness of both the clads at 0.20 mm away from the interface was almost the same, although the bending plastic strain in the HCV4 specimen was larger than that in the HCV2 specimen. In viewing from a different angle, many places with low hardness around 270 HV were found in the HCV2 clads (Figure 10b,c). In contrast, the minimum hardness was around 290 HV at 0.20 mm away from the interface in the HCV4 clad (Figure 10e), although similarly low hardness was observed at 0.25 mm away from the interface in the HCV4 clad (Figure 10f). Such low hardness places were hardly observed at 0.15 mm away from the interface in both the clads. The minimum hardness trend is in good agreement with variation of the bending plastic strain with the Mo thickness of bending specimens in both the clads (Figure 7). The low hardness places would be associated with micro defects such as invisible cracks and voids. The finer rolling texture in the HCV4 clad increased hardness. It resulted in increase of the shock wave damage at the topmost Mo plate of the HCV4 clads (Figure 10f), since the large hardness makes it brittle.

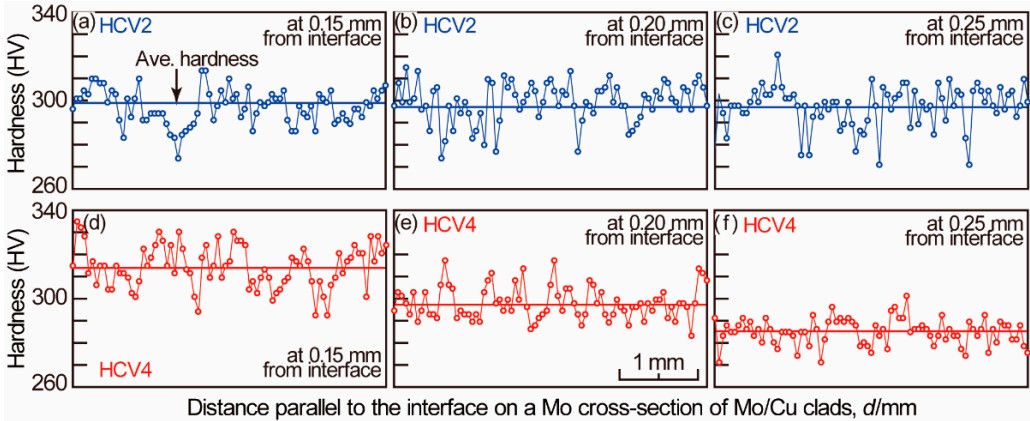

**Figure 10.** Variation of Vickers hardness with distance parallel to the Mo/Cu interface at the three positions away from the interface on a Mo cross section for the as-explosive-welded (**a**–**c**) HCV2 and (**d**–**f**) HCV4 clads obtained at room temperature with an applied load of 0.49 N.

Figure 11 shows a schematic illustration of shock wave damage in a flyer Mo plate of the clads and its effect on bending specimens and behavior for the HCV2 and HCV4 clads. The difference in the bending behavior can be related to the remaining kinetic energy at the interfaces for the HCV2 and HCV4 clads. The kinetic-energy loss at the interface were estimated using the imparted kinetic energy (Figure 2) based on the guide [31], and the remaining kinetic energy could be obtained (Table 4). The remaining kinetic energy at the interface was the highest for the LCV clad, leading to significant high show wave damage in the flyer Mo plate. Similarly, the remaining kinetic energy estimated for the HCV4 clad was a little lower than that for the HCV2 clad. Although the plate velocity for the HCV4 clad was a little higher than that for the HCV2 clad, the imparted kinetic energy of the HCV4 clad became a little lower than that of the HCV2 clad due to thin thickness of a flyer Mo plate. The smaller the remaining kinetic energy is, the smaller the shock wave damage region become. Accordingly, a large part of the flyer Mo plate from the surface had shock wave damage for the HCV2 clad (Figure 11a), in contrast a small part of the flyer Mo plate from the surface had the damage for the HCV4 clad (Figure 11d). When the flyer Mo plate thickness was reduced to 0.20 mm, the flyer Mo plate had a few regions containing shock wave damage (Figure 11b), while the shock wave damage layer was completely removed from the flyer Mo plate for the HCV4 clad (Figure 11e). The difference resulted in difference in fracture of the flyer Mo plate between HCV2 and HCV4 bending specimens (Figure 11c,f). This suggests that removing the shock wave damage region is essential for increasing bending plastic strain. The shock wave damage region was generated near the Mo-plate surface and penetrated through the Mo plate. Decreasing the remaining kinetic energy at the collision interface is essential for decreasing the shock wave damage. This would come true in cladding at high horizontal collision velocity while maintaining a lower collision angle, furthermore with decreasing thickness of a flyer plate.

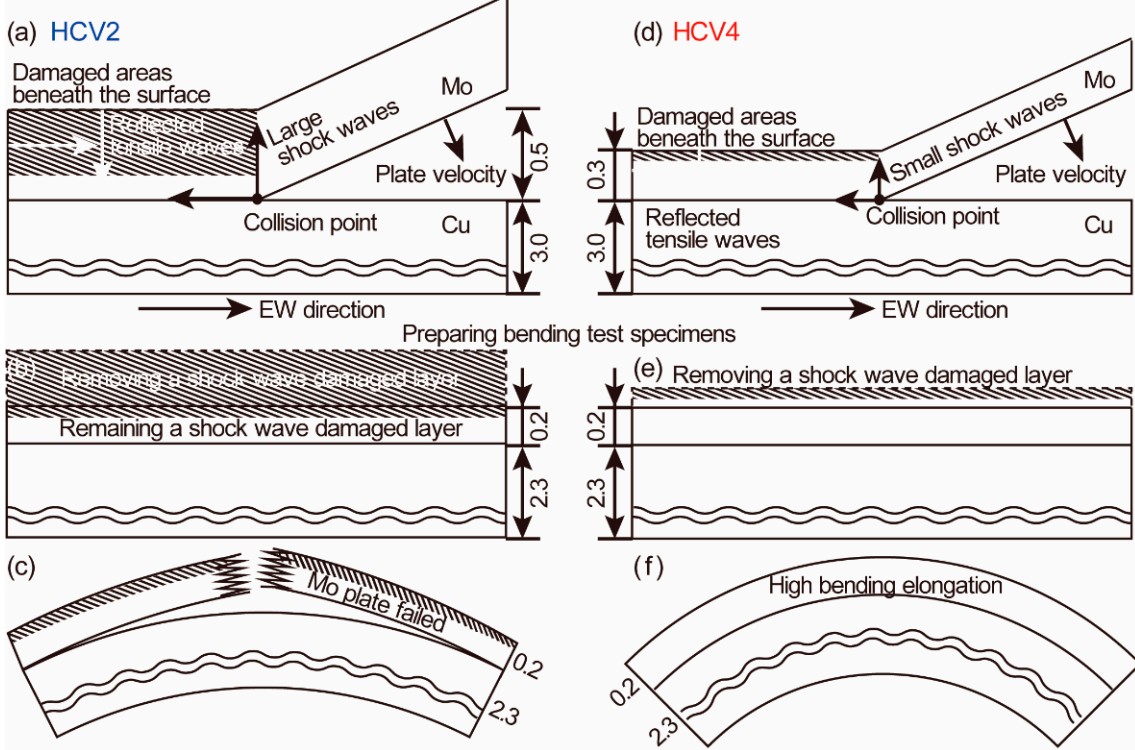

**Figure 11.** Schematic illustrations of (**a**,**d**) shock and reflected-tensile wave initiation during explosive welding and damaged areas beneath the Mo-plate surface, (**b**,**e**) preparation of bending specimens with a 0.25 mm-thick Mo plate, and (**c**,**f**) bending deformation of specimens and failure of a Mo plate for one specimen for HCV2 and HCV4 clads.

**Table 4.** Estimated kinetic energy (KE) imparted to a flyer Mo plate in the explosive welded clads and related values.

| Codes | Plate Velocity (m/s) | Imparted KE (KJ/m$^2$) | KE Loss at the Interface (KJ/m$^2$) | Remaining KE at the Interface (KJ/m$^2$) |
|---|---|---|---|---|
| LCV | 413.61 | 573.324 | 459.308 | 114.656 |
| HCV2 | 270.80 | 245.974 | 196.838 | 49.136 |
| HCV4 | 301.30 | 211.780 | 180.456 | 31.324 |

### 4. Conclusions

This article investigated shock wave damage in the flyer Mo plate of as-explosive-welded Mo/Cu clads and is aimed to get a guideline to reduce it. This study demonstrated dependence of the shock wave damage on horizontal collision velocity and a collision angle in explosive welding, focusing on bending elongation of the clad specimens. Furthermore, influence of the intermediate-layer formation on reducing the damage was characterized. The investigation draws the following conclusions:

(i) Cladding of a flyer Mo plate on a Cu plate succeeded under both the LCV and HCV conditions within the weldability window.

(ii) The HCV clads exhibited larger bending elongation than the LCV clads. Crack was initiated at the surface of the Mo plate, and penetrated through the Mo plate and then propagated along the wavy interface without failure of the Cu plate. This suggests that the HCV condition can reduce the shock wave damage in the Mo plate.

(iii) Micro defects such as cracks and voids were found at the topmost Mo plate of the clads. This indicates shock wave damage produced during the explosive welding. The visible and invisible defects in the Mo plate decreased bending elongation of the clad specimens.

(iv) The shock wave damage was enhanced at the topmost Mo plate with higher hardness in the HCV4 clad, while the damage region was narrower since the remaining kinetic energy at the interface was smaller than that in the HCV2 clad. The bending elongation of HCV2 and HCV4 specimens increased with removing the region with the shock wave damage. The shock wave damage regions depended on the remaining kinetic energy at the collision interface estimated based on the plate velocity, the imparted kinetic energy, and kinetic-energy loss at the interface.

(v) Heat generation due to the intermediate-layer formation at the HCV conditions leaded to loss of the imparted kinetic energy, and the related remaining kinetic energy at the collision interface depended on the Mo-plate thickness as well.

**Author Contributions:** Writing—original draft preparation, conceptualization, and data curation, P.K.P.; writing—review, editing, and project administration, K.I.; investigation and data curation, S.K. and H.Y.; simulation and data curation, A.M.; methodology and resources, S.T. and K.H. All authors have read and agreed to the published version of the manuscript.

**Funding:** This research received no external funding.

**Institutional Review Board Statement:** Not applicable.

**Informed Consent Statement:** Not applicable.

**Data Availability Statement:** Data is contained within the article or supplementary material.

**Acknowledgments:** This work was performed under the Joint Usage/Research Center on Joining and Welding, Osaka University, and Joint Use/Joint Research on Institute of Pulsed Power Science, Kumamoto University.

**Conflicts of Interest:** The authors declare no conflict of interest.

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
