# Peer review of "Characterization of Shock Wave Damages in Explosion Welded Mo/Cu Clads"

_metals, doi:10.3390/met11030501_

Round 1

Reviewer 1 Report

The authors focused on the characteristics of shock wave damage and the identification of the influence of the explosive welding condition on the shock wave damage. The explosive welding technique has been used for many years to join non-weldable metals, so it is not a new technology, but its importance is undisputed. The results of the research have cognitive and application value and are valuable for people involved in the explosive welding technique. The description of the experiment is correct and the results are mostly well documented and interpreted. The material documenting the research results, i.e. drawings and charts, are valuable and correlate with the description in the text of the article. After minor corrections and updating of the literature, the work is eligible for publication in the Metals journal.

Comments and suggestions: 

- The description of the current state of knowledge in the "Introduction" was based on important, but relatively old studies, mostly from the beginning of the 21st century. Of the 15 works cited in the Introduction”, only 3 have been published in the last 5 years. Reading the introduction and analyzing the articles cited in it, the reader may get the impression that the issue of the work is not current or has no cognitive or application potential. And of course it is not. In order to eliminate this impression, the authors should refer to more recent literature in the “Introduction”.

- The authors should provide the full name of the explosive used, i.e. Ammonium Nitrate Fuel Oil. ANFO abbreviation is not understood by everyone.

- Fig. 1: The abbreviation S.O.D should be explained in the text that precedes the figure 1, not in the figure. Moreover, a uniform form of description in the figure should be used, i.e. either S.O.D or “Stand-off distance”. Figure 1a contains both an abbreviation and an explanation of the abbreviation, and fig. 1b only the abbreviation S.O.D itself.

- Fig. 1b is a visualization and compilation of many machining variants, unfortunately the drawing is too complex and not readable. Understanding Figure 1b requires a thorough analysis of the preceding text and the explanations contained therein regarding the color markings used. I suggest simplifying the drawing or, which seems to be the best solution in this case, replacing Figure 1b with a few simplified drawings describing the process parameters in individual variants. An additional explanatory description in the caption under the figure could also be an acceptable form.

- In item 2 "Experimental Details" in line 133 the authors write that "Vickers hardness tests were carried out on the Mo cross sections of clads at RT with an applied load of 0.098 N", then in line 137 they write that "Vickers hardness tests were conducted on the Mo cross sections of clads at RT with an applied load of 0.49 N ”. The reason for using two different loads in hardness measurements should be explained.

- How to interpret the unit [d / mm] in the caption of the horizontal axis in figures 6 and 10

- What load value was used for the hardness measurements in the case of the results shown in Figure 10?

- In the work, the authors use an excessive and unjustified number of abbreviations, which makes reading very difficult, especially since abbreviations are also used to denote treatment variants and research methods. For example, the abbreviation RT, meaning "room temperature", the abbreviation IL, meaning "intermediate layer" and the abbreviation KE , meaning “kinetic energy” are not justified in the text. In lines 176, 225, 277, 279, 292 the use of the abbreviation "EW" is not appropriate. Use the full name "Explosive Welding".

- Lack of explanation in the text of the abbreviation "OM".

Author Response

To Reviewer 1,

Thank you very much for many valuable comments. I revised the text and corresponding figures. We respond about the reviewers’ comments in red as following.

The authors focused on the characteristics of shock wave damage and the identification of the influence of the explosive welding condition on the shock wave damage. The explosive welding technique has been used for many years to join non-weldable metals, so it is not a new technology, but its importance is undisputed. The results of the research have cognitive and application value and are valuable for people involved in the explosive welding technique. The description of the experiment is correct and the results are mostly well documented and interpreted. The material documenting the research results, i.e. drawings and charts, are valuable and correlate with the description in the text of the article. After minor corrections and updating of the literature, the work is eligible for publication in the Metals journal.

Comments and suggestions:

- The description of the current state of knowledge in the "Introduction" was based on important, but relatively old studies, mostly from the beginning of the 21st century. Of the 15 works cited in the Introduction”, only 3 have been published in the last 5 years. Reading the introduction and analyzing the articles cited in it, the reader may get the impression that the issue of the work is not current or has no cognitive or application potential. And of course it is not. In order to eliminate this impression, the authors should refer to more recent literature in the “Introduction”.

-> We add 5 more papers around 2020 and 2021, that reported explosive welding of combinations between brittle materials and Cu, including underwater explosive welding.

- The authors should provide the full name of the explosive used, i.e. Ammonium Nitrate Fuel Oil. ANFO abbreviation is not understood by everyone.

-> The full name of the explosive is provided (in red)

- Fig. 1: The abbreviation S.O.D should be explained in the text that precedes the figure 1, not in the figure. Moreover, a uniform form of description in the figure should be used, i.e. either S.O.D or “Stand-off distance”. Figure 1a contains both an abbreviation and an explanation of the abbreviation, and fig. 1b only the abbreviation S.O.D itself.

-> The full name of “stand-off distance” is described in the text. “S.O.D” is used in Fig. 1a and 1b and the abbreviation S.O.D is explained in the caption.

- Fig. 1b is a visualization and compilation of many machining variants, unfortunately the drawing is too complex and not readable. Understanding Figure 1b requires a thorough analysis of the preceding text and the explanations contained therein regarding the color markings used. I suggest simplifying the drawing or, which seems to be the best solution in this case, replacing Figure 1b with a few simplified drawings describing the process parameters in individual variants. An additional explanatory description in the caption under the figure could also be an acceptable form.

->Fig. 1b is simplified and replaced, and setup parameters are shown in Table 2 that is newly added. The abbreviations S.O.D and PMMA are explained in the caption, together with description of “setup parameters for HCV1-4 clads are shown in Table 2 (newly added)”.

- In item 2 "Experimental Details" in line 133 the authors write that "Vickers hardness tests were carried out on the Mo cross sections of clads at RT with an applied load of 0.098 N", then in line 137 they write that "Vickers hardness tests were conducted on the Mo cross sections of clads at RT with an applied load of 0.49 N ”. The reason for using two different loads in hardness measurements should be explained.

->I add the sentence of “to obtain relatively large apparent contact area probably including invisible micro defects formed by the shock wave damage” after the load of 0.49 N, and similarly add in the 4th paragraph of the chapter 3.3.

- How to interpret the unit [d / mm] in the caption of the horizontal axis in figures 6 and 10

->Usually, the units of the values in x and y axes are different, and thus physical quantity is divided by the unit, leading to a dimensionless quantity. Some of societies recommends the notation. I do not revise them in Figs. 6 and 10.

- What load value was used for the hardness measurements in the case of the results shown in Figure 10?

-> 0.49 N. I add the sentence of “obtained at room temperature with an applied load of 0.49 N” at the end of caption.

- In the work, the authors use an excessive and unjustified number of abbreviations, which makes reading very difficult, especially since abbreviations are also used to denote treatment variants and research methods. For example, the abbreviation RT, meaning "room temperature", the abbreviation IL, meaning "intermediate layer" and the abbreviation KE , meaning “kinetic energy” are not justified in the text. In lines 176, 225, 277, 279, 292 the use of the abbreviation "EW" is not appropriate. Use the full name "Explosive Welding".

-> we deleted abbreviations, and their full names are described.

- Lack of explanation in the text of the abbreviation "OM".

-> we added an optical microscope (OM) in the 3rd paragraph in “Experimental Details”.

Reviewer 2 Report

Manuscript ID: metals-1147843

Title: Characterization of Shock Wave Damages in Explosion Welded Mo/Cu Clads

Authors: Pradeep K Parchuri, Shota Kotegawa, Kazuhiro Ito *, Hajime Yamamoto,

Akihisa Mori, Shigeru Tanaka, Kazuyuki Hokamoto

The paper investigates explosion welding of molybdenum and copper. The authors use different configurations to produce the best quality joints. The main quality criterion in this paper is chosen as the shockwave damage. The topic of explosion welding is certainly relevant and will be of interest to readers. The material is also interesting, difficult to weld. All other things aside, the staging of the experiment is standard and not unique. In general, the work is performed at a high level, but there are comments:

  1. The idea of bend testing itself raises a number of questions. Of course, a damaged stretched sheet will fracture. However, the non-fracture may also indicate a tear in the joint and a displacement of the layer. Such a possibility is not excluded in the work. Fractured samples are not analyzed in any way except visually.
  2. Comparison of HCV2 and HCV4. Your results show that the grains in the HCV4 sample are more refined. This may, in particular, be responsible for the greater bending resistance. This is confirmed by microhardness measurements (Figure 10). In the conclusions, however, this is not discussed and the entire effect is attributed to defects. As a result, it may well be that the damage from the shock wave contributes a negligible effect.
  3. It is not clear where exactly the images from Figure 9 were obtained from. Thus, so far, there is no proof of any damage at all.
  4. On the mechanical test plots, the strain axis must be scaled. The plots must start from the origin of the coordinates.
  5. 29% self-referencing is a bit much. Journals usually stick to the 15% limit. Metals Journal does not set a hard limit, but I recommend following common ethics.

Author Response

To Reviewer 2,

Thank you very much for many valuable comments. I revised the text and corresponding figures. We respond about the reviewers’ comments in red as following.

The paper investigates explosion welding of molybdenum and copper. The authors use different configurations to produce the best quality joints. The main quality criterion in this paper is chosen as the shockwave damage. The topic of explosion welding is certainly relevant and will be of interest to readers. The material is also interesting, difficult to weld. All other things aside, the staging of the experiment is standard and not unique. In general, the work is performed at a high level, but there are comments:

The idea of bend testing itself raises a number of questions. Of course, a damaged stretched sheet will fracture. However, the non-fracture may also indicate a tear in the joint and a displacement of the layer. Such a possibility is not excluded in the work. Fractured samples are not analyzed in any way except visually.

->No, only OM observation of the specimens after bending tests. You are definitely correct. The analysis of a Mo stretched sheet joined on a Cu plate after bending tests was afraid to be too much for this study. A tear in the Mo sheet in the joints (clads) is difficult to find visually, I guess even after bending tests. Thus, we conducted indentation tests in Mo-sheet cross sections of the as-explosion-welded clads with relatively large load to obtain large apparent contact area probably including invisible micro defects. The three-point bending tests (load concentrated on the topmost Mo sheet in the center of bending specimens), we believe, could be visualized the difference of shock wave damage that accumulated near Mo-sheet surface.

Comparison of HCV2 and HCV4. Your results show that the grains in the HCV4 sample are more refined. This may, in particular, be responsible for the greater bending resistance. This is confirmed by microhardness measurements (Figure 10). In the conclusions, however, this is not discussed and the entire effect is attributed to defects. As a result, it may well be that the damage from the shock wave contributes a negligible effect.

->Yes, you are right. That was caused by thinner thickness of a flyer Mo plate to control the imparted kinetic energy. Although the maximum hardness was observed in the HCV4 clad, the bending strength was always higher in the HCV2 clad than the HCV4 clad. I add the related description in the corresponding positions of the text as follows:

“The lateral elongated rolling texture was finer in the HCV4 clad than that in the HCV2 clad, since the Mo-plate thickness of the HCV4 clad was thinner than that of the HCV2 clad (Table 2).” in the latter half of the 1st paragraph of the chapter 3.2, “The maximum hardness was higher in the HCV4 clad (about 315 HV) than in the HCV2 clad (300 HV). The higher hardness in the HCV4 clad is associated with the finer rolling texture.” at the end of the 2nd paragraph of the chapter 3.2, “… , although the finer rolling texture and the higher maximum hardness in the HCV4 clads was observed (Figures 5 and 6)” in the first half of the 1st paragraph of the chapter 3.3, “The finer rolling texture in the HCV4 clad increased hardness. It resulted in increase of the shock wave damage at the topmost Mo plate of the HCV4 clads (Figure 10(f)), since the large hardness makes it brittle.” at the end of the 4th paragraph of the chapter 3.3, “The shock wave damage was enhanced at the topmost Mo plate with higher hardness in the HCV4 clad, while the damage region was narrower since the remaining kinetic energy at the interface was smaller than that in the HCV2 clad.” at (iv) in Conclusions.

It is not clear where exactly the images from Figure 9 were obtained from. Thus, so far, there is no proof of any damage at all.

->We said in the text in the third paragraph of chapter 3.3 below Figure 10: “… micro cracks and voids around grain boundaries were found in the CROSS-SECTIONAL SIM images obtained at the TOPMOST REGION of the flyer Mo plates …”, but “cross-sectional” and “topmost” confused. I revised “… found in the cross-sectional SIM images perpendicular to the Mo-plate surface obtained underneath the surface of the flyer Mo plates …”.

On the mechanical test plots, the strain axis must be scaled. The plots must start from the origin of the coordinates.

->I am very sorry for that. Figure 4(a) was revised that the plots start from the origin of the coordinates. On the other hand, please leave Fig. 7 as it is, to be easy to compare the strength and elongation between HCV2 and HCV4 clads as a function of a Mo-plate thickness.

29% self-referencing is a bit much. Journals usually stick to the 15% limit. Metals Journal does not set a hard limit, but I recommend following common ethics.

->I am very sorry for that. References numbers of 26 and 27 are deleted and leave no. 28 to explain how to calculate the kinetic energy, etc. in table 3 (old numbering). Furthermore, we add five more references published in 2020 and 2021 in introduction.

Reviewer 3 Report

Article: Characterization of Shock Wave Damages in Explosion Welded Mo/Cu Clads

The title is consistent with the content of the paper.

Introduction:

Summarizes recent research related to the topic sufficiently. The need for research is objectively justified.

Experimental Details:

The initial state and properties of the materials used are not presented. What is used Molybdenum total elongation? 5…20%?  The EW test arrangements are comprehensively presented.

Three point bending test speed and temperature has been told. Also fulcrum distances. What is used punch tool diameter?

How the strain is measured in bending test?

Results and Discussions:

Figure 3. FE-SEM image a) has a different magnification than the other pictures, why?

What is Mo total elongation after EW process in different cases? Does the bending test test more for material elongation than joint damages?

My suggestion is that you supplement the text by explaining what proportion the material properties (yield strength, tensile strength, tensile strength) play in a three-point bending test like the one used.

Conclusions: Ok.

Author Response

To Reviewer 3,

Thank you very much for many valuable comments. I revised the text and corresponding figures. We respond about the reviewers’ comments in red as following.

The title is consistent with the content of the paper.

Introduction:

Summarizes recent research related to the topic sufficiently. The need for research is objectively justified.

Experimental Details:

The initial state and properties of the materials used are not presented. What is used Molybdenum total elongation? 5…20%?  The EW test arrangements are comprehensively presented.

->We did not conduct a bending test of as-received Mo plates from Niraco Corp. We checked the mechanical properties in the Niraco Corp site, but only hardness of 259-263 HV is available. Sorry, we cannot add the data since the plate was totally used for EW.

Three point bending test speed and temperature has been told. Also fulcrum distances. What is used punch tool diameter?

->6 mm in diameter. I add the tool diameter in the text, together with how to calculate the fulcrum distance in the table 3 caption.

How the strain is measured in bending test?

->the bending strain of specimens was not directly measured, but it was estimated from the crosshead displacement in the tensile testing machine (Shimadzu Corp.) based on the conversion equation … . I add the explanation in the 4th paragraph of Experimental Details.

Results and Discussions:

Figure 3. FE-SEM image a) has a different magnification than the other pictures, why?

->Figure 3a was obtained from a LCV clad specimen, while others were from HCV clad specimens. The wave period of an undulant interface was longer in the LCV clads than that in the HCV clads. I add the explanation in the 1st paragraph of Results and Discussions.

What is Mo total elongation after EW process in different cases? Does the bending test more for material elongation than joint damages?

->You can see Figure 8. No crack or fracture of a Mo plate in the Mo/Cu clads was observed in the specimens with thin Mo plates. It suggests that the bending elongation of a Mo plate without shock wave damage can be more than that of a Mo plate with shock wave damage. Unfortunately, we do not know elongation of the Mo plate itself, and we cannot be estimated joint advantage and disadvantage in the Mo plate of the clads. We revised the end of the 1st paragraph of the chapter 3.3; “Noted that bending strain of the clad specimens was associated with whether the damaged layer in the Mo plate of the as-explosive-welded clads could be removed or not. In addition, the shock wave damage was accumulated near the Mo plate surface of the as-explosive-welded clads than near the interface.” (revision in red).

My suggestion is that you supplement the text by explaining what proportion the material properties (yield strength, tensile strength, tensile strength) play in a three-point bending test like the one used.

->It is somehow a difficult option or mandatory comment. I add the equations for estimating the bending stress and strain from load indicated by a load cell and crosshead displacement. Usually, the coefficient is larger than the case of tensile/compressive tests, and thus the bending stress is larger than tensile/compressive stress, while strain is a little bit complicated. We add the explanation at the end of the 4th paragraph of Experimental Details.

Conclusions: Ok.

Round 2

Reviewer 2 Report

To reliably separate the effects of flying sheet damage from the effects of sheet adhesion, I suggest that next time you do a pull-off or shear test.

Reviewer 3 Report

Thanks for the improvements. I have no further suggestions for improvement.